# Prominin-1 Regulates Retinal Pigment Epithelium Homeostasis: Transcriptomic Insights into Degenerative Mechanisms

**DOI:** 10.3390/ijms262311539

**Published:** 2025-11-28

**Authors:** Weihong Huo, Jinggang Yin, Purnima Ghose, Jenny C. Schafer, Edward Chaum, Sujoy Bhattacharya

**Affiliations:** 1Departments of Ophthalmology and Visual Sciences, Vanderbilt University Medical Center, Nashville, TN 37232, USA; weihong.huo.2@vanderbilt.edu (W.H.); jinggangyin@gmail.com (J.Y.); purnima.ghose@vumc.org (P.G.); edward.chaum@vumc.org (E.C.); 2Cell and Developmental Biology, Vanderbilt University, Nashville, TN 37232, USA; jenny.c.schafer@vanderbilt.edu

**Keywords:** inherited retinal dystrophies, atrophic age-related macular degeneration, epithelial–mesenchymal transition, *mTORC1*, MerTk, *PINK1*, Gremlin-1, phagocytosis, autophagy, proliferation, bile acids

## Abstract

Inherited retinal degenerations (IRDs), driven by pathogenic mutations, often involve primary dysfunction of the retinal pigment epithelium (RPE)—a pathogenic feature shared with atrophic age-related macular degeneration (aAMD), despite aAMD’s multifactorial etiology. *Prominin-1* (*Prom1*), traditionally linked to photoreceptor pathology, has an unclear role in RPE homeostasis. We assessed Prom1 expression in C57BL/6J mouse retina sections and RPE flat mounts using immunohistochemistry and generated *Prom1*-knockout (KO) mouse RPE cells via CRISPR/Cas9. Bulk RNA sequencing with DESeq2 and gene set enrichment analysis (GSEA) revealed *Prom1*-regulated pathways. *Prom1*-KO cells exhibited upregulation of *Grem1*, *Slc7a11*, *Serpine2*, *Il1r1*, and *IL33* and downregulation of *Ablim1*, *Cldn2*, *IGFBP-2*, *BMP3*, and *OGN*. Hallmark pathway interrogation identified reduced expression of *PINK1* (mitophagy) and *MerTK* (phagocytosis), implicating defects in mitochondrial quality control and outer segment clearance. Enrichment analysis revealed activation of E2F/MYC targets, mTORC1 signaling, oxidative phosphorylation, and TNFα/NF-κB signaling, alongside suppression of apical junctions, bile acid metabolism, and Epithelial-Mesenchymal Transition (EMT) pathways. These findings suggest *Prom1* safeguards RPE integrity by modulating stress responses, mitochondrial turnover, phagocytosis, metabolism, and junctional stability. Our study uncovers *Prom1*-dependent signaling networks, providing mechanistic insights into RPE degeneration relevant to both IRD and aAMD, and highlights potential therapeutic targets for preserving retinal health.

## 1. Introduction

Inherited retinal dystrophies (IRDs) are a genetically heterogeneous group of disorders that cause progressive vision loss and, in many cases, irreversible blindness [1,2]. While many IRDs have traditionally been viewed as photoreceptor-centric diseases, increasing evidence suggests that retinal pigment epithelium (RPE) dysfunction plays a critical, and possibly primary, role in disease progression, particularly in IRDs, in which the mutated gene is present in both RPE and photoreceptors [3,4]. This distinction is particularly relevant for IRDs associated with *Prominin-1* (*Prom1*, also known as CD133), where clinical features such as RPE atrophy and photoreceptor degeneration closely resemble those seen in atrophic age-related macular degeneration (aAMD) [5,6]. Despite this phenotypic overlap, the mechanisms by which *Prom1* regulates RPE homeostasis remain poorly understood, representing a significant gap in our understanding of macular disease pathogenesis.

*Prom1* is a pentaspan transmembrane glycoprotein widely recognized as a stem cell and cancer stem cell marker [7,8]. Beyond its role in stemness, *Prom1* is expressed in differentiated epithelial and non-epithelial cells [9], glial cells [9], and the adult retina [10], suggesting broader physiological functions. In photoreceptors, *Prom1* localizes to the base of the outer segments, where it regulates disk morphogenesis and membrane architecture [11,12]. Loss-of-function mutations in *Prom1* cause a spectrum of retinal diseases, including autosomal dominant and recessive retinitis pigmentosa [11,13], cone-rod dystrophies [14,15,16], and macular dystrophies such as Stargardt disease type 4 (STGD4) [12,17].

While *Prom1*-related retinal dystrophies have traditionally been viewed as photoreceptor-centric, emerging evidence suggests a complex pathophysiology involving RPE. In a *Prom1*-null *Xenopus laevis* model, CRISPR/Cas9-mediated Prom1 loss led to age-dependent RPE degeneration and subretinal drusenoid-like deposits that preceded photoreceptor loss, challenging the paradigm that Prom1 dysfunction primarily affects photoreceptors [18]. These deposits resembled human drusenoid material, suggesting that RPE pathology is an initiating event in Prom1-associated degeneration.

Stargardt disease 4 (STGD4) shares clinical and pathological features with ABCA4-related Stargardt disease type 1 (STGD1) and the atrophic form of AMD, including central photoreceptor degeneration and RPE atrophy [19,20,21]. Although STGD1 is driven by bisretinoid lipofuscin accumulation due to ABCA4 dysfunction, STGD4 may involve overlapping or distinct mechanisms. Notably, *Prom1* mutations such as p.R373C and c.869delG have been associated with parafoveal RPE atrophy, granular mottling, and thinning of the outer retina in patients [5,22,23]. In younger individuals, spectral-domain OCT reveals early RPE/Bruch’s membrane thinning and progression to geographic atrophy (GA) [24]. Longitudinal studies confirm the expansion of GA, profound outer retinal degeneration, and phenotypic variability in patients with the Prom1 R373C mutation [25]. These findings suggest that *Prom1* dysfunction may directly impair RPE integrity, contributing to macular degeneration independent of bisretinoid accumulation.

The relationship between *Prom1* and *ABCA4* is further supported by studies showing additive effects of *Prom1* and *Abca4* mutations on RPE pathology, including granular mottling and lipofuscin-like deposits [19,22]. Despite these clinical observations, the mechanistic role of *Prom1* in RPE biology remains poorly defined. Our own studies have shown that *Prom1* is expressed in human RPE and localizes predominantly to the cytoplasm, where it regulates autophagy by modulating mTORC1/2 signaling and autophagosome trafficking [26]. Additionally, we confirmed *Prom1* expression in mouse RPE (mRPE) in situ by immunogold electron microscopy and RNAscope assays [27]. We showed that targeted Prom1 knockdown in situ using AAV2/1 vectors induces RPE cell death and photoreceptor degeneration, recapitulating the features of aAMD [27].

We recently demonstrated that *Prom1* loss in mRPE activates mTORC1, suppresses TFEB activity, and induces epithelial–mesenchymal transition (EMT), implicating *Prom1*-mTORC1-TFEB signaling as a central regulator of RPE homeostasis [28]. Although *Prom1*-related IRDs have long been considered photoreceptor cell-autonomous pathologies, emerging evidence suggests that *Prom1* is a major driver of RPE integrity and function. To address this knowledge gap, we performed transcriptomic profiling of *Prom1*-deficient mRPE cells to identify molecular pathways disrupted by *Prom1* loss. Using bulk RNA sequencing and gene set enrichment analysis (GSEA), we identified dysregulated genes and pathways involved in stress signaling, autophagy, lysosomal function, and epithelial identity—molecular signatures that mirror RPE dysfunction in IRDs and aAMD. These findings support a model in which *Prom1* maintains RPE homeostasis in a cell-autonomous manner, suggesting that its loss contributes to retinal degeneration through mechanisms beyond photoreceptor disk morphogenesis.

## 2. Results

### 2.1. Prom1 Is Expressed in Mouse RPE In Situ

We recently showed Prom1 mRNA localization in mouse RPE in situ using RNAscope, as well as Prom1 protein localization in mitochondria of mouse RPE, by immunogold electron microscopy [27]. To confirm Prom1 protein expression in mRPE, we performed Immunohistochemistry (IHC) on mouse RPE flat mounts. Our studies show Prom1 staining (green) alongside ZO-1 (red), a tight junction marker in mouse RPE flat mounts. At 40× magnification, Prom1 was predominantly localized in the RPE cytoplasm, with enrichment in nuclear and perinuclear regions and partial association with apical junctions marked by ZO-1 (Figure 1). At 40× zoom, this pattern was more pronounced, with Prom1 puncta evident near cell borders, cytoplasm, and concentrated around nuclei. The merged images show partial co-localization of Prom1 with ZO-1, which supports its weak association with apical domains (Figure 1). The presence of Prom1 signal in mRPE cytoplasmic and nuclear compartments suggests potential roles in trafficking or signaling beyond membrane organization. Consistent with these observations, Prom1 was detected (green) in the RPE layer in situ in mouse retinal sections (white arrowheads), evidenced by 20×, 40×, and 63× confocal images, confirming its expression outside photoreceptors and providing protein-level evidence of its presence and subcellular organization, but Prom1 staining was not observed in the negative control (Appendix B showing z-stack 3D-orthogonal images of multiple retinal layers processed as negative control) (Figure 2). These observations are consistent with our previous findings, which showed Prom1 mRNA expression by RNAscope in the mouse RPE, the inner segment (IS) of photoreceptors, as well as a small group of cells in the inner retina [27]. The 3D orthogonal reconstruction shows that the Prom1 protein is present within the RPE monolayer, spatially distinct from photoreceptor outer segments. The labeling pattern in the XZ/YZ plane appears punctate cytoplasmic with apical enrichment, consistent with intracellular localization in RPE rather than a junctional or membrane-restricted signal (Figure 2A). Western blotting analysis of mRPE cells cultured in vitro confirmed robust Prom1 expression in wild-type mRPE and its absence in Prom1-knockout (KO) samples (using Clustered Regulatory Interspaced Short Palindromic Repeats (CRISPR)/CRISPR-associated protein 9 (Cas9) (Figure 2B). At the same time, qPCR demonstrated significant Prom1 transcript reduction in Prom1-KO cells (Figure 2C). Together, these results establish Prom1 as a cytoplasmic and junction-associated protein in mRPE, supporting its potential role in maintaining RPE homeostasis.

### 2.2. Bulk RNA Sequencing of WT and Prom1-KO mRPE Cells

To investigate the molecular consequences of Prom1 loss in retinal pigment epithelium (RPE), we performed bulk RNA sequencing on wild-type (WT) and Prom1-knockout (KO) mouse RPE cells. Sequencing reads were normalized for differences in sequencing depth and filtered to remove lowly or non-expressed genes, ensuring that downstream analyses focused on biologically relevant transcripts. Genes with fewer than five counts in at least three samples were excluded to minimize noise from sporadically expressed features. Raw counts, normalized counts, and post-filtering values were visualized on a log2 + 1 scale to include zero-count features (Figure 3). As expected, increasing sequencing depth improved feature capture, consistent with coverage principles for bulk RNA-seq. After normalization and filtering, WT and KO samples exhibited comparable distributions, confirming effective correction for sequencing depth and removal of low-abundance genes (Figure 3). This approach provided a robust dataset for differential expression analysis and pathway enrichment, enabling the identification of Prom1-dependent signaling networks.

Principal component analysis (PCA) was performed to visualize variance across WT and *Prom1*-KO mRPE transcriptomes. Uncorrected PCA revealed clustering by sequencing batch, indicating a strong batch effect (Appendix A). After applying batch correction with limma, PC1 accounted for 92% of the variance, while PC2 accounted for the residual variability of unknown origin. Importantly, batch correction improved sample clustering by genotype, confirming that biological differences rather than technical artifacts drive the major variance component (Appendix A). Two Prom1-KO samples (8995-SB-3 and 9262-SB-6) remained outliers after correction, suggesting intrinsic biological heterogeneity or technical anomalies (Appendix A). These findings validate the need for batch adjustment and confirm that genotype is the dominant source of variance in the dataset. Following the removal of two outlier Prom1-KO samples (8995-SB-3 and 9262-SB-6), PCA demonstrated improved clustering by genotype (Appendix A). PC1 accounted for 96% of the total variance, indicating that the primary source of variability was the experimental condition rather than technical factors. WT and Prom1-KO samples separated clearly along PC1, confirming robust transcriptional differences between groups. PC2 explained only 3% of the variance and did not correspond to batch or condition, suggesting minimal residual confounding (Appendix A). These findings validate the dataset’s data quality and support its suitability for downstream differential expression and pathway analyses.

### 2.3. Differential Expression and Gene Set Enrichment Analyses

Differential expression analysis revealed a pronounced transcriptional shift, including 15 upregulated and downregulated genes (Figure 4A). The volcano plot summarizing DESeq2 data for WT and *Prom1*-KO mRPE cells shows a cluster of high-magnitude, high-significance hits on both tails, with prominent upregulated genes with a positive log2 fold change (red), including *Gremlin-1* (*Grem1*), an endogenous BMP antagonist that induces EMT in fetal RPE cells [29]; *Serpine 2*, a serpin with neurotropic and anti-angiogenic activities secreted by the RPE into the interphotoreceptor matrix toward the neural retina [30]; *Interleukin-receptor-like 1*(*Il1r1*), an activator of the complement alternative pathway in RPE cells in macular degeneration [31]; *Interleukin 33* (*IL33*), a key regulator of inflammation and RPE atrophy in AMD [32]; and *retinoic acid-induced 14* (*Rai14*), which is expressed in the RPE and has been shown to promote mTOR-mediated inflammation [33,34] (Figure 4A). The downregulated genes with a negative log2 fold change (blue) include *actin-binding LIM protein family member 1* (*Ablim1*), which directly binds F-actin to serve as a scaffold for signaling modules of the actin cytoskeleton and governs the formation of dense cortical actin meshwork to prevent mechanical tension-induced blebbing during hTERT-RPE1 migration [35,36]; *IGF binding protein 2* (*IGFBP2*), which plays a significant role in modulating IGF-1 activity in the RPE, and its loss induces an early reactive RPE phenotype [37,38]; *Bone morphogenetic protein-3* (*BMP3*), its reduced expression leads to fibrosis; and *Osteoglycin* (*OGN*), a proteoglycan with strong expression in the Bruch’s membrane and choroid [39] (Figure 4A). We used gene set enrichment analysis (GSEA) to rank gene lists and identify enriched pathways (Figure 4B). Enrichment scores were calculated by hypergeometric tests and normalized to the size of the gene set. A positive normalized gene enrichment score (NES) indicates that the gene set is upregulated in the experimental group. Preliminary analysis of hallmark gene sets from the Molecular Signature Database shows significant upregulation of hallmark signatures for cell cycle transcription factors (*E2F* and *MYC* targets, *G2M* checkpoint), *mTORC1* signaling, unfolded protein response (UPR), reactive oxygen species (ROS) pathway, TNFA signaling via NF-kappaB, DNA repair, and oxidative phosphorylation (Figure 4B). Downregulated pathways of interest include apical junction, bile acid metabolism, and EMT pathways (Figure 4B). Together, the DEG landscape and Hallmark Pathway analysis indicate a shift toward RPE innate immune/stress response remodeling, accompanied by dampening of mitochondrial energy metabolism and cell cycle progression, providing pathway-level context for the gene-wise changes observed due to the loss of *Prom1*.

### 2.4. Heatmap Analysis and Biological Context

The heatmap of the top 50 differentially expressed genes (sorted by *p*-value) shows strong clustering of biological replicates, confirming data reproducibility and condition-specific transcriptional signatures (Figure 5). A separate heatmap of the top 100 DEGs further confirms major transcriptional differences between ET and Prom1-KO mRPE cells (Appendix A). *Prom1*-KO samples (9262-SB-5, 9262-SB-4, and 8995-SB-2) cluster distinctly from WT samples, reflecting robust genotype-driven expression differences (Figure 5). Several genes with known or emerging roles in RPE physiology and stress response are prominently represented. For example, MertK, essential for photoreceptor outer segment phagocytosis, is reduced in *Prom1*-KO mRPE compared to WT, suggesting impaired RPE clearance and homeostasis [40]. *Cldn2*, which is involved in tight junction integrity in murine RPE, is downregulated in *Prom1*-KO mRPE, suggesting potential disruption of barrier properties [41]. *Tgf-beta* receptor 2 (*Tgfbr2*) maintains RPE barrier integrity, and its reduction suggests loss of homeostatic signaling [42]. Caveolin-1 (*Cav1*), a scaffolding protein in RPE caveolae that regulates vesicular trafficking and endocytosis, is upregulated, suggesting altered lipid raft dynamics that amplify stress signaling [43]. *Map3k5* activates JNK and MAPK pathways under oxidative stress, and its upregulation in *Prom1*-KO mRPE suggests heightened stress signaling and apoptosis susceptibility [44]. Metabolic regulators, such as *SLC38A1* and *SLC7A11* (cysteine/glutamate transporter), are upregulated in *Prom1*-KO mRPE, reflecting shifts in energy and redox balance that align with the suppression of oxidative phosphorylation and lipid metabolism [45]. Upregulation of *Il33* and downregulation of *Igfbp2* further support the activation of innate immunity/inflammatory pathways and downregulation of metabolic signaling, echoing the enrichment of the *TNFα/NF-κB* and *IL6–JAK–STAT3* pathways. Collectively, these gene-level changes support the interpretation that *Prom1* loss drives a transition from homeostatic RPE function toward a stress- and inflammatory-related phenotype with metabolic compromise.

### 2.5. Gene Network Map in Prom1-KO Versus WT mRPE

Network mapping of *Prom1-KO* versus *WT* mouse RPE transcriptomes identified five highly connected hallmark clusters—*MYC targets*, *E2F targets*, *G2M checkpoint*, and *mTORC1 signaling*—each reflecting coordinated programs of RPE stress, metabolic reprogramming, and degeneration (Figure 6). The *mTORC1* cluster, including *Serp1*, *Map2k3*, *Slc2a1*, and *Slc7a11*, indicates enhanced nutrient sensing and stress signaling consistent with mTORC1 hyperactivation and impaired autophagic flux. Within the G2M checkpoint cluster, *Aurka*, *Ccnb2*, *Plk1*, and *Slc38a1* are included, indicating altered amino acid transport and mitotic checkpoint activation. The MYC and E2F target cluster includes *Bub1b*, *Cdk1*, *Npm1*, *Mcm5*, *Mcm7*, *Hmgb3*, and *Hmgb2*, which are associated with glycolytic shift, chromatin remodeling, and proliferative stress (Figure 6). Collectively, these cluster-specific changes reinforce a model where *Prom1* loss drives inflammatory signaling, metabolic stress, and impaired autophagy, converging on pathways implicated in RPE dysfunction and retinal disease.

### 2.6. Validation of Transcriptomic Data

While our work showed earlier that *Prom1* loss induces EMT in mRPE cells, GSEA in the current study reveals a reduction in Hallmark EMT pathway activation in Prom1-KO mRPE cells [28]. To further investigate this, we performed gene-level interrogation of EMT-related transcripts using stringent differential expression criteria (Log2FC > 1 and adjusted *p*-value ≤ 0.05), visualized in a volcano plot that highlights the Hallmark EMT gene set. The volcano plot revealed significant upregulation of EMT-associated genes, including *Grem1*, *Pcolce2*, *Cxcl2*, *CD44*, and *Serpine2*, while *IGFBP2 and POSTN* were notably downregulated (Figure 7A). qPCR confirmed *IGFBP2* and *POSTN* gene downregulation (Figure 7B,C) and elevated expression of *Grem1* gene in *Prom1*-deficient RPE cells (KO22 and KO26.4) (Figure 7D). Western blot confirmed increased Grem1 protein in *Prom1*-KO cells; however, densitometric analysis showed a statistically significant increase only in KO26.4, while KO22 exhibited a non-significant (ns) trend (Figure 7E). These findings suggest that *Prom1* loss upregulates *Grem1* transcription, but protein levels may be influenced by post-transcriptional regulation or by the specific genome region disrupted within the *Prom1* gene in each CRISPR-edited *Prom1*-KO clone (Figure 7E). Although GSEA did not reveal clear enrichment for EMT-related pathways, targeted analysis of individual gene expression changes provides evidence for EMT-like alterations, which is consistent with our earlier observations showing EMT in *Prom1*-KO mRPE cells [28]. The downregulation of extracellular matrix remodeling (*POSTN*) and the upregulation of an EMT-promoting gene (*Grem1*), coupled with the suppression of epithelial-supporting factors (*IGFBP2*), suggest that *Prom1* deficiency may initiate a partial or context-specific EMT program that is not fully captured by canonical pathway-level analyses.

To define how *Prom1* loss perturbs RPE function, we examined the DEGs highlighted in the volcano plot across oxidative stress/reactive oxygen species (ROS), unfolded protein response (UPR), phagocytosis, and *TNF/NF-kappaB* pathways, using stringent differential expression criteria (Log2FC > 1, adjusted *p*-value ≥ 0.05) (Figure 8A). Among ROS-related genes, *Egfr* was significantly downregulated, consistent with prior work showing that oxidative stress reduces *Egfr/Erk-Akt* signaling and impairs RPE viability; conversely, *Egfr*-mediated antioxidant rescue restores RPE survival [46,47]. *Fibulin-5* (*Fbln5*) was also decreased; *Fbln5* is an ECM protein that limits integrin-driven ROS generation and modulates redox tone—its loss is linked to heightened oxidative stress in vivo, suggesting reduced ECM-based ROS control in the RPE microenvironment [48]. *Pax2* downregulation is noteworthy given its established role as redundant with *Pax6* in specifying RPE fate during development; reduced *Pax2* is therefore compatible with impaired RPE identity/homeostasis under stress [49]. Finally, *PINK1* gene expression was significantly reduced, suggesting defective mitophagy in the RPE. *PINK1* loss drives mitochondrial *ROS-Nrf2* signaling, EMT-like changes, and structural abnormalities, all of which can contribute to RPE degeneration [50]. qPCR confirmed *PINK1* was significantly reduced in *Prom1*-KO mRPE groups, indicating impaired mitochondrial quality control due to loss of *Prom1* (Figure 8B).

In contrast, several oxidative-response genes were upregulated (Figure 8A). *Nqo1*, a canonical *Nrf2* target that blunts redox cycling, was increased, consistent with a compensatory antioxidant response to elevated ROS [51]. *Map3k5* (*ASK1*), a redox-sensitive *Map3k* that activates *JNK/p38* under oxidative stress, was elevated, aligning with pro-apoptotic and pro-inflammatory stress signaling in stressed epithelia [52]. *Hgf* was increased; *Hgf* signaling in RPE promotes proliferation/motility and disrupts tight/adherens junctions via *MET/Akt*, indicating junctional remodeling and RPE barrier compromise under stress [53,54]. *Ripk3* was also upregulated; *RIPK1/3* activity has been implicated in RPE injury by promoting mitochondrial DNA release, *AIM2* inflammasome activation, and inflammatory cell death, supporting a necroptosis-prone state in diseased RPE [55]. *Sod3* gene levels increased as well; *SOD3* protects the ECM from superoxide, and its modulation strongly influences retinal function in vivo, consistent with an extracellular antioxidant countermeasure to oxidative burden [56]. UPR markers such as *Qrich1*, a transcriptional effector driving proteostasis under ER stress, and *Ptpn2*, which protects epithelia from ER-stress-induced death, were upregulated, suggesting activation of UPR signaling in stressed mRPE [57,58]. Finally, the *TNF/NF-kB* pathway component was increased in *Prom1-KO* mRPE. *Traf1* is an *NF-kB*-inducible adapter that modulates inflammatory and pro-survival signaling and is a classic footprint of active *TNF/NF-kB* circuits [59]. Together, these changes map a coordinated phenotype in *Prom1*-deficient mRPE—elevated oxidative and ER stress, impaired mitochondrial quality control, junctional/ECM remodeling, and TNF-driven inflammation.

The volcano plot revealed downregulation of the *PINK1* gene in *Prom1*-KO cells (Figure 8A). qPCR confirmed a significant reduction in *PINK1* gene expression, indicating impaired mitochondrial quality control and mitophagy signaling in *Prom1*-deficient mRPE cells (Figure 8B). Of note, the *MerTk* gene was downregulated (Figure 8A). Consistent with transcriptomic findings, Western blotting showed a significant reduction in the MerTK protein in both *Prom1*-KO22 and -KO26.4 mRPE cells, and qPCR confirmed reduced MerTK gene expression in these cells (Figure 8C,D). Because MerTk is indispensable for daily outer segment clearance by RPE, reduced MerTK expression provides a mechanistic link to impaired phagocytic flux and secondary photoreceptor stress [60].

qPCR studies confirmed the upregulation of the *SLC7A11* gene, involved in RPE oxidative stress response (Figure 8E) [61]; and downregulation of genes, including cytoskeletal remodeling, *Ablim1* (Figure 8F) [35]; extracellular matrix organization, *OGN* (Figure 8G); and retinoic acid synthesis, *Aldh1a1* (Figure 8H) [62], in *Prom1*-KO mRPE cells. These findings show that *Prom1* deficiency disrupts multiple pathways critical for RPE homeostasis, including mitochondrial turnover, phagocytosis, oxidative stress regulation, and ECM integrity—mechanisms that may underlie retinal degeneration in *Prom1*-associated disease.

## 3. Discussion

*Prom1* has long been recognized as a primary driver of cell-autonomous photoreceptor pathology, yet its role in the retinal pigment epithelium (RPE) has largely remained unexplored. Building on our prior observations of *Prom1* transcripts in mRPE, this study provides definitive evidence of *Prom1* protein expression in mRPE both in situ and in cultured cells, confirming its presence beyond photoreceptors [27]. Although *Prom1* expression in the RPE monolayer is quantitatively lower than in the photoreceptor-rich retina, our data demonstrate that *Prom1* is not merely incidental; instead, it functions as a major regulator of RPE homeostasis in a cell-autonomous manner [27,28]. Through transcriptomic analysis, we show that *Prom1* loss triggers a cascade of stress, metabolic, and structural disruptions that converge on pathways central to RPE integrity. These findings reposition *Prom1* from a photoreceptor-centric protein to a critical node in RPE biology, with implications for both IRDs and aAMD. Loss of *Prom1* reprograms mouse RPE cells toward a degenerative state marked by activation of stress and inflammatory molecules (*TNF/NF-kB*; *Map3k5*, *RIPK3*), failure of mitochondrial quality control (reduced *PINK1*), reduced phagocytic capacity (reduced *MerTk*), junctional/ECM remodeling compatible with *HGF* signaling, and broader metabolic rewiring (including *mTORC1* activation, *SLC7A11* induction, and suppression of bile acid metabolism) together with transcriptional features of incomplete EMT—an array of changes that closely mirror mechanisms implicated in both IRDs and aAMD [63,64].

Although *Prom1* is a pentaspan membrane protein, its limited co-localization with *ZO-1* in mRPE aligns with its known enrichment in cholesterol-rich, highly curved apical microdomains such as microvilli and cilia, regions that are spatially separate from the RPE tight junctions marked by *ZO-1*. This microdomain specificity, observed across epithelial and sensory cells, indicates that *Prom1*’s role involves membrane curvature and protrusion dynamics rather than functioning as a junctional scaffold [9,65]. The prominent cytoplasmic and nuclear localization of *Prom1* in mRPE in situ can be explained by several factors: (1) active vesicular trafficking and endosomal recycling, including its presence in microvillus-derived vesicles and intracellular vesicles termed intercellsomes [66]; (2) glycosylation-dependent epitope accessibility, where intracellular pools in the endoplasmic reticulum, Golgi, and mitochondria may be more likely to be detected due to altered glycosylation patterns [65]; and (3) rare but documented nuclear trafficking in specific cell types, indicating biological plausibility for nuclear *Prom1* signals [67]. Importantly, our published data confirm that this cytoplasmic distribution pattern is conserved in human RPE donor explants (using immunocytochemistry) and mouse RPE mitochondria in situ (using immunogold electron microscopy), further supporting the biological significance of these mechanisms, particularly as a cytoplasmic regulator of essential RPE functions such as autophagy [26,27].

Loss of *Prom1* disrupts multiple RPE survival programs through interconnected mechanisms. Increased mTORC1 signaling in *Prom1*-KO RPE correlates with impaired RPE autophagy and lysosomal activity, which drive EMT and result in AMD-like pathology [28,68]. Reduced *PINK1* compromises mitophagy, allowing damaged mitochondria to accumulate, which elevates ROS and activates retrograde stress signaling that promotes EMT-like changes [69]. In parallel, downregulation of *MERTK* impairs phagocytosis of photoreceptor outer segments, a process essential for retinal homeostasis; its failure accelerates photoreceptor stress and degeneration [60]. These vulnerabilities are compounded by upregulation of *ASK1/MAP3K5* and *RIPK3*, which amplify oxidative and ER stress responses, driving *JNK/p38*-mediated apoptosis and necroptosis [55]. Junctional instability observed in *Prom1*-KO cells is consistent with *HGF/MET* signaling, which is known to dismantle tight junctions, increase RPE motility, and activate *AKT* survival pathways [54]. Finally, the induction of *SLC7A11* reflects an adaptive antioxidant response that counters ferroptosis, underscoring a shift toward stress tolerance rather than the restoration of homeostasis [70]. Together, these changes form a coherent degenerative program that links mitochondrial failure, impaired clearance, inflammatory signaling, and barrier breakdown —hallmarks of RPE dysfunction in AMD and IRDs.

Among our most intriguing findings is the robust suppression of the Hallmark Bile Acid metabolism pathway in *Prom1*-KO mRPE cells. Bile acids and their receptors (*FXR* and *TGR5*) are traditionally viewed from a hepatic–intestinal perspective; however, they also function as hormone-like immunometabolic signals that help limit inflammation, maintain epithelial barrier integrity, and regulate energy metabolism across various tissues [71]. Emerging evidence suggests that bile acid receptors in the retina/RPE function as potent immunometabolic regulators of inflammation, oxidative stress, epithelial barrier integrity, lipid homeostasis, and mitochondrial function. Several mechanisms could underlie the suppressed bile acid pathway signature in our model: (1) Loss of epithelial polarity and microdomain architecture (microvilli, membrane curvature) due to Prom1 deficiency likely perturbs bile acid receptor/transporter trafficking, thereby blunting its signaling. (2) Mitochondrial and lysosomal stress, including reduced *PINK1* and *mTORC1* activation caused by the loss of Prom1, which can be mitigated or worsened by the *FXR/TGR5* signaling axis due to their roles in mitochondrial biogenesis, quality control, and lipid metabolism [72]. Therefore, impaired bile acid metabolism can be both a cause and a consequence of mitochondrial dysfunction. (3) Inflammatory crosstalk, where reduced bile acid signaling removes an anti-inflammatory brake, potentially amplifying *TNF-NF-kappaB* signaling in *Prom1*-KO mRPE cells. (4) Gut–retina systemic–metabolic axis, as gut-derived bile acids can reach the eye and modulate oxidative/mitochondrial stress, suggesting altered metabolism or transport, could further sensitize *Prom1*-deficient RPE [73]. *TUDCA/UDCA* mitigates oxidative and ER stress, dampens pro-inflammatory cytokines, stabilizes barrier properties, and promotes autophagy-mediated cytoprotection in RPE, suggesting that loss of bile acid signaling removes an endogenous pro-homeostatic brake in *Prom1*-deficient RPE and offers an actionable therapeutic axis [74,75]. Reduced *MerTK*-dependent phagocytosis and junctional/ECM remodeling increase structural strain, which bile acid signaling typically mitigates through blood–retinal barrier support and *FXR*-mediated gene regulation [76,77]. The induction of *SLC7A11* and ferroptosis-adaptive programs in *Prom1*-KO mRPE cells further suggests that bile acids can counter lipid peroxidative stress through lipid detoxification and crosstalk between peroxisomes and mitochondria. Our data may be among the first to link *Prom1* loss to bile acids, highlighting therapeutic implications: restoring bile acid signaling via receptor agonists or strategies that augment endogenous bile acid metabolism could stabilize *Prom1*-deficient RPE and slow degeneration. These metabolite-signaling considerations are consistent with broader lipid metabolic dysfunction observed in multi-omics RPE models and aAMD patient serum, reinforcing the clinical relevance of our metabolic signatures [78].

The global architecture of the *Prom1*-KO transcriptome aligns with previous RPE transcriptomics studies in human AMD [63,64]. Combined meta-analyses and single-cell atlases highlight inflammatory and proteostatic remodeling, as well as metabolic rewiring (including TNFα/NF-κB activation, UPR/proteostasis engagement, and changes in oxidative metabolism) in AMD regions, paralleling the pathway enrichments observed in *Prom1*-KO mRPE cells. Longitudinal RNA-seq of RPE during outer segment processing documents coordinated autophagy, lysosome, and proteostasis programs, aligning with our UPR signatures and supporting a central role for proteostasis failure in disease progression [72,79]. Genetic activation of mTOR in mouse RPE induces EMT, disrupts autophagy, drives metabolic imbalance, and causes aAMD-like pathology [68]. This mirrors the mTORC1/EMT transcriptomic changes in *Prom1*-KO mRPE, supporting the idea that metabolism and cell-state transitions are closely linked in degenerating RPE [68]. Furthermore, our combination of EMT-effector induction with overall suppression of the EMT hallmark aligns with the emerging concept of incomplete or partial EMT in AMD and RPE injury—loss of epithelial features with partial mesenchymal acquisition rather than full transdifferentiation, reconciling transcriptomic complexity with pathology [80].

Our transcriptomic data from *Prom1*-KO mRPE cells reveal a paradoxical pattern: individual canonical EMT-promoting genes (such as *GREM1*) are upregulated, while Claudin family (*Cldn1* and *Cldn2*) genes are downregulated, and *Lhx2* is downregulated (Appendix A). These changes indicate defects in the outer blood–retinal barrier and downregulation of the RPE differentiation program, leading to a less functional RPE. Yet, the global Hallmark EMT gene set is suppressed (using GSEA). At first glance, this seems contradictory. However, growing evidence in both epithelial biology and RPE research suggests that EMT is not a simple epithelial–mesenchymal switch but rather a continuous spectrum of cell states, ranging from fully epithelial to hybrid or partial EMT, and ultimately to fully mesenchymal [81,82]. In the context of the RPE, a “partial EMT” phenotype is gaining attention, in which cells may lose epithelial polarity, begin to express mesenchymal markers, or acquire migratory potential, while retaining some epithelial features [80,83]. In Prom1-KO mRPE cells, the upregulation of EMT-effector genes indicates the start of a phenotypic switch, but the overall suppression of the canonical EMT signature suggests that the complete mesenchymal program is not engaged. Therefore, we interpret this as a stalled or incomplete EMT—a state in which the cells embark on the transition path but remain locked in an intermediate state [80]. The partial EMT concept is highly relevant in degenerative RPE diseases (such as aAMD) rather than the classical complete EMT seen in fibrosis or cancer. In aAMD, there is evidence for RPE cells acquiring migratory or ECM remodeling characteristics while retaining epithelial identity [81]. We propose that Prom1 loss shifts cultured RPE cells toward a destabilized epithelial state by dismantling the epithelial program (i.e., reduced junctional proteins, ZO-1, and Claudins) but does not fully activate the mesenchymal signature (hence GSEA suppression). A partial EMT explains why *Prom1*-KO RPE cells exhibit junctional/ECM remodeling yet do not fully adopt a fibroblast-like phenotype, consistent with an RPE cell continuum model in which cells move from a stable epithelial state toward mesenchymal phenotypes without completing differentiation. This hybrid state can be pathological because RPE cells lose their strict epithelial features but remain in place, leading to chronic degeneration rather than acute fibrosis. We therefore argue that the suppression of the Hallmark EMT gene set should not be seen as a lack of transitional change but rather as a sign that the program is incomplete, consistent with the concept of hybrid EMT states in RPE. Future research will be needed to understand the functional effects of partial EMT versus full-EMT states of the RPE in their natural location.

### Limitations

Our study has several limitations. First, while transcriptomic profiling provides a comprehensive review of gene expression changes in Prom1-KO mRPE cells, it does not capture post-transcriptional regulation, protein turnover, or dynamic signaling events, all of which may profoundly influence disease phenotypes. Secondly, our study utilizes bulk RNA sequencing of *Prom1*-KO mRPE cells cultured in vitro, which does not fully reflect the in vivo heterogeneity and spatial organization of the RPE in the central or peripheral regions. The in vitro system lacks the native tissue architecture and microenvironmental cues that influence RPE subpopulation dynamics, including partial EMT and stress adaptation. Third, although we integrate prior localization studies confirming that Prom1 is a cytoplasmic protein in both human and mouse RPE, our localization data in this study do not directly link the localization pattern to functional outcomes such as vesicular trafficking or signaling compartmentalization. Finally, while transcriptomic analyses provide valuable insights into primary-level perturbations in Prom1-KO mRPE cells, they reflect correlative changes in gene expression and do not fully account for dynamic regulatory mechanisms at the protein level. To particularly address this, we included protein-level validation for select markers such as *MerTk* and *GREM1*; however, comprehensive protein profiling was limited by the lack of commercially available antibodies for several key targets. Future studies integrating proteomic approaches may help refine mechanistic interpretations related to mitochondrial quality, EMT, autophagy, and junctional integrity. The development of an RPE-specific inducible Prom1-KO mouse model would be particularly valuable, as it would enable in vivo testing of whether Prom1 loss alone drives mitochondrial failure, autophagy impairment, junctional instability, partial EMT, and cell-autonomous RPE degeneration through similar transcriptomic changes—phenocopying key pathogenic mechanisms shared between aAMD and IRDs. In parallel, single-cell or spatial transcriptomic approaches applied to intact retinal tissue lacking Prom1 will be essential to validate and extend our findings, as well as to delineate context-dependent transcriptional responses across RPE subtypes within their native microenvironment.

## 4. Materials and Methods

### 4.1. Reagents

Materials purchased include the following: Fetal bovine serum (FBS, R&D Systems, Minneapolis, MN, USA); Enhanced chemiluminescence (ECL) Western blot detection system (Perkin Elmer, Inc., Shelton, CT, USA); Protease/Phosphatase Inhibitor Cocktail (Thermo Fisher Scientific, Waltham, MA, USA); *Prom1/CD133* rabbit polyclonal antibody (OAA100379, Aviva Systems Biology, Corp., San Diego, CA, USA), *Prom1* rabbit polyclonal (abcam, Waltham, MA, USA, ab19898); MertK (Invitrogen, Carlsbad, CA, USA 14-5751-82), ZO-1 monoclonal antibody (Invitrogen, ZO1-1A12, 33-9100) and GREM1 polyclonal antibody (catalog# PA5-119163, Invitrogen), goat anti-rabbit Alexa Fluor 488 (Thermo Fisher, cat#A-11008), donkey anti-mouse Alexa Fluor 647 (Thermo Fisher, cat#A-31571) and *Prom1* gRNA (Thermo Fisher Scientific, Waltham, MA, USA).

### 4.2. Mice and Colony Management

C57/BL6J mice were obtained from the Jackson Laboratory (Bar Harbor, ME, USA) (stock #000664). Mice were housed, maintained on a 12 h light–dark cycle, and provided food and water ad libitum. The Institutional Animal Care and Use Committee of Vanderbilt University Medical Center (VUMC) approved all experiments. All animal procedures followed the guidelines of the Association for Research in Vision and Ophthalmology Statement on the Use of Animals in Ophthalmic and Vision Research. Both male and female mice (6–8 weeks old) were used for this project.

### 4.3. Cell Culture

Mouse RPE (mRPE) cells were obtained from Rosario Fernandez-Godino (Harvard, MA, USA), as described earlier [84]. Briefly, isolated mRPE cells from 8- to 12-week-old C57/BL6J mice were pooled and plated on a 24-well plate coated with laminin (10 mg/mL). After 72 h, cells were allowed to reach 50% confluence and then transduced overnight with Lenti-HPV E6/E7 (10^6^ TU/mL) (Applied Biological Materials, Richmond, BC, Canada cat #G268) in the presence of 4 mg/mL polybrene. Cells were selected in RPE media containing 5% FBS and puromycin (1 mg/mL) for 12 days to generate immortalized mRPE cells. These cells were cultured in a culturing medium containing N1 medium supplement (1/100 vol/vol), glutamine (1/100 vol/vol), penicillin–streptomycin (1/100 vol/vol), non-essential amino acid (1/100 vol/vol), hydrocortisone (20 mg/mL), taurine (250 mg/L), triiodothyronine (0.013 mg/L), 5% FBS in alpha-MEM at 37 °C in 5% CO_2_ and media replaced three times a week. The cells were subcultured using trypsin (0.25%) and frozen in 10% FBS with 10% DMSO.

### 4.4. Generation of Prom1-Deficient mRPE Cells via CRISPR/Cas9

*Prom1* was knocked out in mRPE cells by CRISPR/Cas9-mediated gene editing, as described earlier [28]. mRPE cells were cultured in 6-well plates. The cells were allowed to reach 70% confluency, then transduced with 10 μL of the lentiviral Cas9 construct and 2 μL of polybrene. After 24 h, the infected cells were subcultured into media containing 1.5 mg/mL puromycin and grown to 90% confluency. The cells were selected in media containing 0.5 mg/mL puromycin for an additional week. After puromycin selection, the stably expressing Cas9-mRPE cells were transfected with Prom1 (chromosome 5) synthetic guide RNA (gRNA) sequence (5′-CGTTGCTGCAACAAATGCGG-3′) (ThermoFisher Scientific, catalog # A35533) using Lipofectamine CRISPRMAX transfection reagent (ThermoFisher, Cat # CMAX00008) to target the mouse *Prom1* gene at exon 5, following the manufacturer’s protocol. *Prom1-KO* was verified by genomic DNA analysis. Cas9-expressing mRPE cells were transfected with either scrambled or *Prom1* gRNA, and these cells were then used to extract genomic DNA using the QIAamp DNA Micro Kit. PCR was performed using the forward primer 5′-GTGCATACTGGGGTCCTCAC-3′ and reverse primer 5′-ATCTCCCTGCAACACCCTAA-3′ and Taq PCR master mix kit, Qiagen, Germantown, MD, USA. Genomic sequences were analyzed using Basic Local Alignment Search Tool (BLAST), version 2.15.0, National Library of Medicine, Bethesda, MD, USA, accessed on 15 June 2023 to confirm the WT and different *Prom1*-KO sequences. We introduced two deletions at separate genomic sites within *Prom1*, generating two independent Prom1-KO lines, as described earlier [28].

### 4.5. Mouse RPE Flat Mount Preparation and Immunohistochemistry

After euthanasia, the mouse eyes were enucleated and fixed in neutral buffered formalin for 15 min at room temperature. They were washed twice in PBS, and the anterior segment (cornea and lens) was removed to expose the posterior eyecup. The neural retina was carefully detached, leaving the RPE–choroid–sclera intact. Four radial cuts were made from the periphery toward the optic nerve to flatten the eyecup, as described earlier [85]. The tissue was placed RPE-side up in a dish or on a slide. Samples were permeabilized and blocked in PBS containing 0.5% Triton X-100 and 5% normal donkey serum (Abcam, ab7475) for 1 h. The primary antibody diluted in blocking buffer was applied and incubated overnight at 4 °C, followed by PBS washes and secondary antibody incubation for 2 h at room temperature. Finally, tissues were mounted in Fluoromount-G medium containing 4’,6-diamidino-2-phenylindole (DAPI) (Thermo Fisher, Waltham, MA, USA, 00-4959-52) and imaged using confocal microscopy.

### 4.6. Mouse Retina Sections and Confocal Imaging

The enucleated mouse eyes were immersed in 4% PFA overnight at 4 °C and rinsed 3 times in PBS. The eyes were immersed in 10% sucrose in PBS for 2–4 h, transferred to 20% sucrose, and finally to 30% sucrose overnight at 4 °C. The eyes were embedded in cryomold, sectioned at 10 μm with a cryostat, mounted on slides, and stored at −80 °C. The cryo slides were taken out of the freezer, warmed for 10 min at room temperature, washed twice in PBS with 0.5% Triton X-100 for 15 min, blocked for 2 h at room temperature in 1× PBST with 5% normal donkey serum, incubated with primary antibody overnight in 1× PBS, followed by three washes in 1× PBS for 10 min, and incubated with secondary antibody in 1× PBST with 5% NDS for 1 h at room temperature. Slides were washed in 0.5% TritonX-100 in PBS and coverslipped with Fluoromount-G with DAPI. No primary antibody was used for the negative control slide; however, the slides were incubated with secondary antibody and coverslipped with Fluoromount-G containing DAPI, as described above. Confocal images of retina sections and RPE flatmount samples were captured using a Zeiss LSM880 confocal microscope with Zeiss ZEN (black) 2.3 software, applying the following parameters: For retina section images at 20× magnification: Plan-Apochromat 20×/0.8 objective; 405 nm excitation at 1.5% and 488 nm excitation at 1.0%; detection wavelengths from 410 to 495 nm and 495–630 nm; gain set to 800; offset at 0; pixel dwell time of 1.02 microseconds per pixel; averaged twice; Z-stack of 12 slices covering approximately 25 microns at Nyquist sampling; scale = 0.42 × 0.42 × 0.29 microns per pixel. For retina section images at 40×: Plan-Apochromat 40×/1.3 Oil DIC UV-IR objective; 405 nm excitation at 1.5% and 488 nm excitation at 1.5%; detection wavelengths from 410 to 495 nm and 495–630 nm; gain = 800; offset = 0; pixel dwell time of 1.02 microseconds per pixel; averaged twice; Z-stack of 18 slices over approximately 18 microns at Nyquist sampling; scale = 0.21 × 0.21 × 1.07 microns per pixel. For retina section images at 63×: Plan-Apochromat 63×/1.4 Oil DIC objective; 405 nm excitation at 2.2% and 488 nm excitation at 2.4%; detection wavelengths from 410 to 495 nm and 495–630 nm; gain = 800; offset = 0; pixel dwell time of 1.02 microseconds per pixel; averaged twice; Z-stack of 25 slices over approximately 12 microns at Nyquist sampling; scale = 0.13 × 0.13 × 0.53 microns per pixel. For RPE flatmount images at 40×: Plan-Apochromat 40×/1.3 Oil DIC UV-IR objective; 488 nm excitation at 0.7% and 633 nm excitation at 6.5%; detection wavelengths from 493 to 628 nm and 638–755 nm; gain = 800; offset = 0; pixel dwell time of 2.05 microseconds per pixel; averaged twice; Z-stack of 11 slices over roughly 8 microns at Nyquist sampling; scale = 0.21 × 0.21 × 0.82 microns per pixel. Maximum Intensity Projections and orthogonal projections were generated using Zeiss ZEN (blue) 3.8.

### 4.7. Western Blotting

Cell lysates were prepared using mammalian protein extraction buffer (Cell Signaling Technology, Beverly, MA, USA) and a Halt protease/phosphatase inhibitor cocktail (Thermo Fisher Scientific, Waltham, MA, USA), followed by SDS-PAGE, as described previously [26,86]. Proteins were transferred to Immobilon–PVDF membranes with 0.45 μm pore size (Millipore, Bedford, MA, USA) and incubated overnight with primary antibodies at 4 °C in Tris-buffered saline containing 0.1% Tween-20 and 5% nonfat dry milk (Biorad, Hercules, CA, USA). Membranes were subsequently incubated with horseradish peroxidase-conjugated secondary antibodies at room temperature for 1 h, and the immune complexes were visualized using the ECL detection system (PerkinElmer, Waltham, MA, USA) and the Azure c500 Imaging Biosystem (Dublin, CA, USA). Membranes were stripped and re-probed for actin as a loading control. Representative Western blots for three experiments are shown. Densitometric analysis of all Western blots was performed using ImageJ software 1.47t Java 1.6.0_20 (developed by Wayne Rasband, National Institutes of Health, Bethesda, MD, USA; available at http://rsb.info.nih.gov/ij/index.html (accessed on 20 Aug 2025)), as described earlier [26]. Blots used in the figures comply with the digital image and integrity policies. The uncropped full-length blots are available as a Appendix A.

### 4.8. Real-Time Quantitative PCR

TRIzol reagent (Thermo Fisher Scientific, Waltham, MA, USA) was used to extract total RNA from WT and *Prom1*-KO mRPE cells, as described earlier [28]. Total RNA concentrations were quantified by measuring the A260 and A280 absorbance using a NanoDrop spectrophotometer, as described previously [26]. Total RNA (1 mg) was reverse-transcribed into cDNA using a kit from Promega (Madison, WI, USA) according to the manufacturer’s instructions. The cDNA was diluted 1:5 with DNase-free water. Real-time qPCR was performed using an Ariamx Real-Time PCR system (Agilent Technologies, Santa Clara, CA, USA) with 2.5 ml of the cDNA product in a 25 mL reaction mixture containing 1× SYBR^®^ Green Master Mix (Applied Biosystems, Foster City, CA, USA) and 120 nM forward and reverse primers. A list of forward and reverse primers used for qPCR analyses of genes in WT vs. Prom1-KO mRPE cells is provided in Table 1.

### 4.9. Bulk RNA Sequencing and Data Analysis

Total RNA was isolated from retinal pigment epithelium (RPE) cells dissected from wild-type (WT) and *Prom1*-knockout (KO) mouse RPE using the RNeasy Mini Kit (Qiagen, Germantown, MD, USA) according to the manufacturer’s protocol. RNA integrity was assessed using the Agilent 2100 Bioanalyzer, and samples with an RNA Integrity Number (RIN) of 8.0 or higher were selected for sequencing to ensure high-quality input. Bulk RNA sequencing was performed by the Vanderbilt Technologies for Advanced Genomics (VANTAGE) core facility at Vanderbilt University Medical Center. Libraries were prepared using the Illumina TruSeq Stranded mRNA Library Prep Kit, and paired-end sequencing (150 bp reads) was conducted on the Illumina NovaSeq 6000 platform. Raw sequencing reads were trimmed for adapter sequences and filtered for quality using *TrimGalore* v0.6.7 (https://zenodo.org/records/5127899 (accessed on 20 January 2023)) by Creative Data Solutions, Vanderbilt University. High-quality reads were aligned to the *Mus musculus* reference genome (mm39) using *STAR* v2.7.9a [87] with the quantMode GeneCounts parameter enabled to generate gene-level count matrices. Gene-level counts were normalized and analyzed for differential gene expression (DEG) using *DESeq2* v1.36.0 [88]. Genes with fewer than five counts across at least three samples were excluded to reduce background noise. Differentially expressed genes were identified using an adjusted *p*-value threshold of 0.05 and a log2 fold change > 1.2. Gene set enrichment analysis (GSEA) was performed using Hallmark Gene sets from the Molecular Signatures Database (MSigDB v7.5) (https://www.gsea-msigdb.org/gsea/msigdb/ (accessed on 20 January 2023)). Enriched gene sets were visualized and clustered in Cytoscape (v3.9.1) using the EnrichmentMap and AutoAnnotate plug-ins to generate functional network representations.

### 4.10. Statistical Analysis

All data were analyzed using GraphPad Prism 9 (GraphPad Software, Inc., San Diego, CA, USA). Data are expressed as mean ± SE. Experiments were repeated three times, with triplicate samples for each. An unpaired 2-tailed Student’s *t*-test and Bonferroni post hoc testing were used to assess statistical significance. Unless otherwise stated, * *p* < 0.05, ** *p* < 0.01, *** *p* < 0.001, and **** *p* < 0.0001 values were considered significant; ns, not significant.

## 5. Conclusions

In summary, our study positions RPE-localized *Prom1* at a central point linking autophagy and mitophagy with mitochondrial quality control, phagocytic activity, epithelial barrier integrity, and immunometabolic signaling. The *Prom1*-dependent network we identify not only aligns with human RPE transcriptomics in AMD and IRDs but also provides a targeted set of mechanistic, actionable entry points for maintaining RPE health across genetic and age-related retinal degeneration.

## Figures and Tables

**Figure 1 ijms-26-11539-f001:**
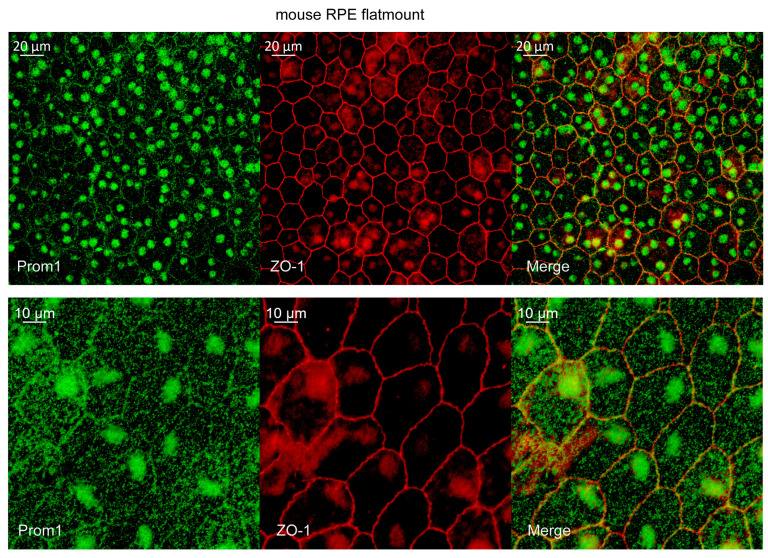
Prom1 is expressed in both mRPE in situ. Immunohistochemistry of WT mouse RPE flatmounts with *Prom1* (green) and *ZO-1* (red) antibodies. Confocal images with 40× objective and 40× zoom.

**Figure 2 ijms-26-11539-f002:**
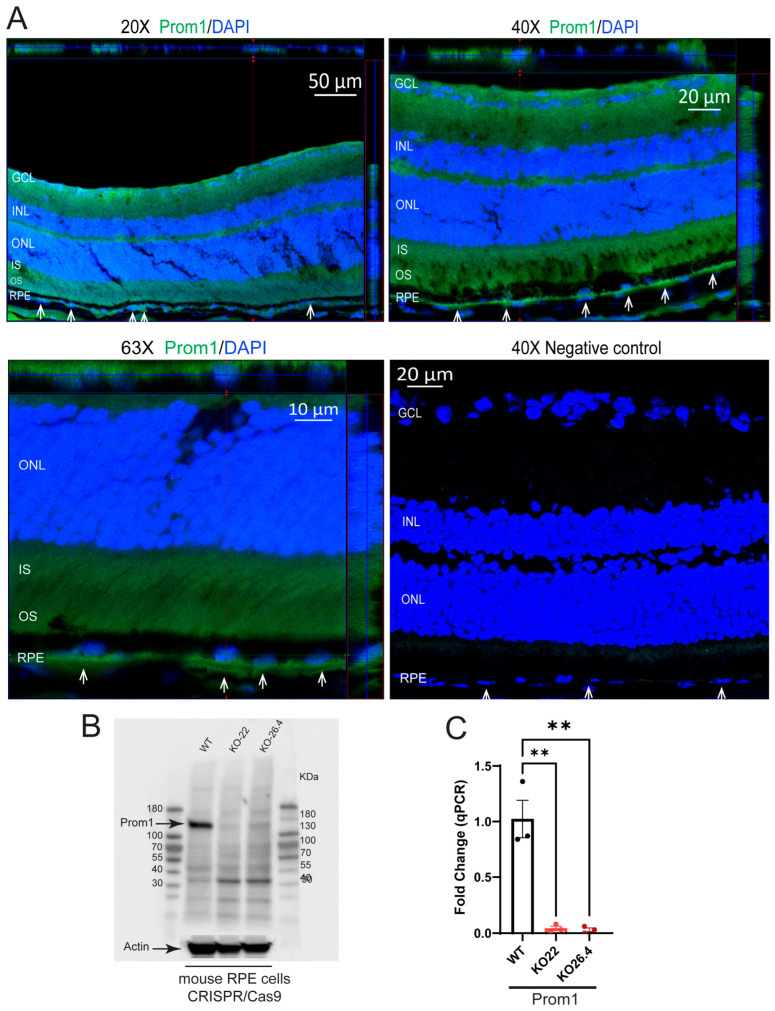
Prom1 expression in mouse retina sections and mouse RPE cells. (**A**) WT mouse retina sections were stained for *Prom1* (green) and DAPI (blue) at 20×, 40×, and 63× objectives with 3D orthogonal views (white arrowheads showing *Prom1* expression in RPE), and no *Prom1* labeling in the negative control at 40×. (**B**) Western blot analysis showing *Prom1* protein levels in WT and *Prom1*-KO (KO-22 and KO-26.4) cells. (**C**) Relative *Prom1* gene expression in WT, *Prom1*-KO22, and KO26.4 cells. WT compared to KO-22 (** *p* < 0.01) and WT compared to KO-26.4 (** *p* < 0.01).

**Figure 3 ijms-26-11539-f003:**
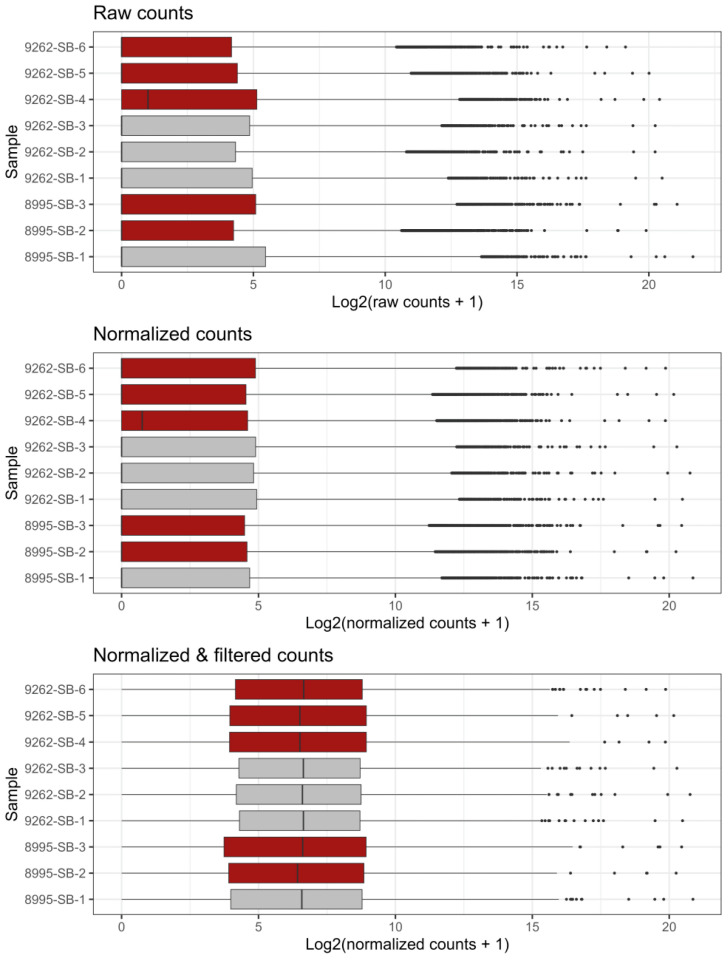
RNA sequencing counts, normalized and filtered. The number of raw sequencing counts before normalization, post-normalization, and post-filtering on a Log2 + 1 scale to visualize features with 0 counts. Counts for each gene were normalized to the sequencing depth for each sample. Genes were filtered out if they were counted fewer than 5 times across at least three samples. WT samples are in grey (9262-SB-1, -2, -3, and 8995-SB-1), and Prom1-KO samples (9262-SB-4, -5, -6, and 8995-SB-1, -2) are in red.

**Figure 4 ijms-26-11539-f004:**
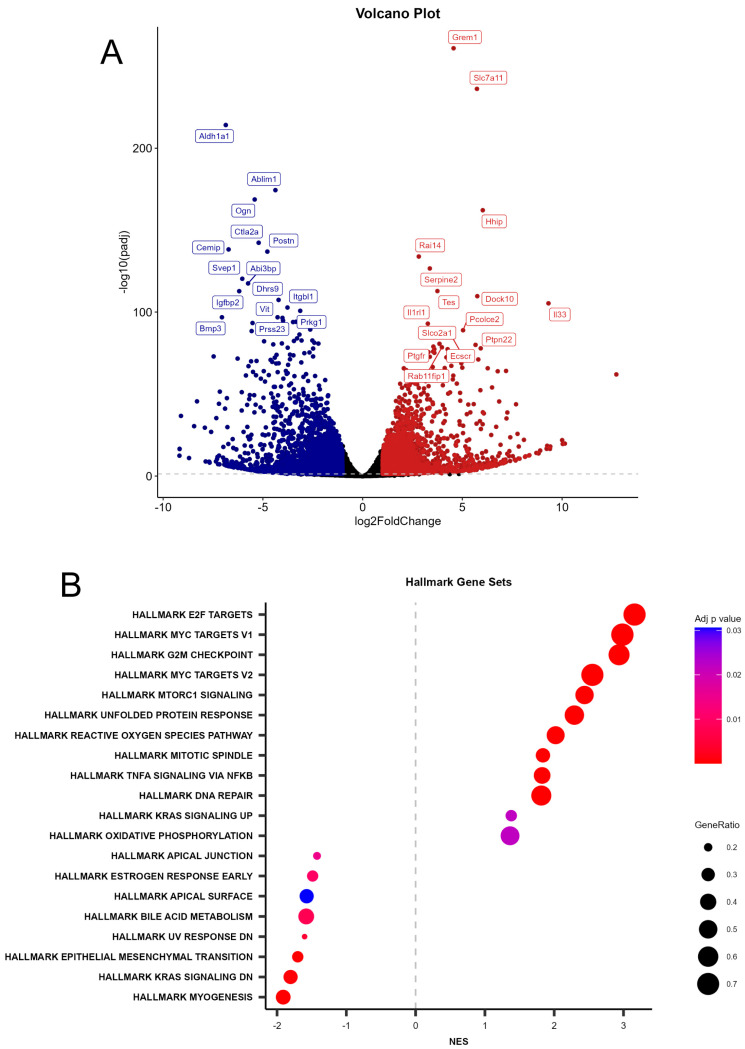
Differential expression of genes and gene enrichment analysis in WT vs. Prom1-KO mRPE cells. (**A**) Volcano plot illustrating the top 15 differentially expressed genes (DEGs) in *Prom1-KO vs. WT* mRPE cells by bulk RNA sequencing. Significantly upregulated (in red) and downregulated (in blue) are labeled; n = 3/group. Selected notable genes are labeled for reference. (**B**) The normalized gene enrichment score (NES) of Hallmark gene sets from the Molecular Signature Database reveals up- and downregulated pathways in *Prom1-KO* vs. *WT* mRPE cells (n = 3/group). Dot size corresponds to the number of genes in each set, while color reflects adjusted *p*-value with a gradient from blue (less significant) to red (highly significant).

**Figure 5 ijms-26-11539-f005:**
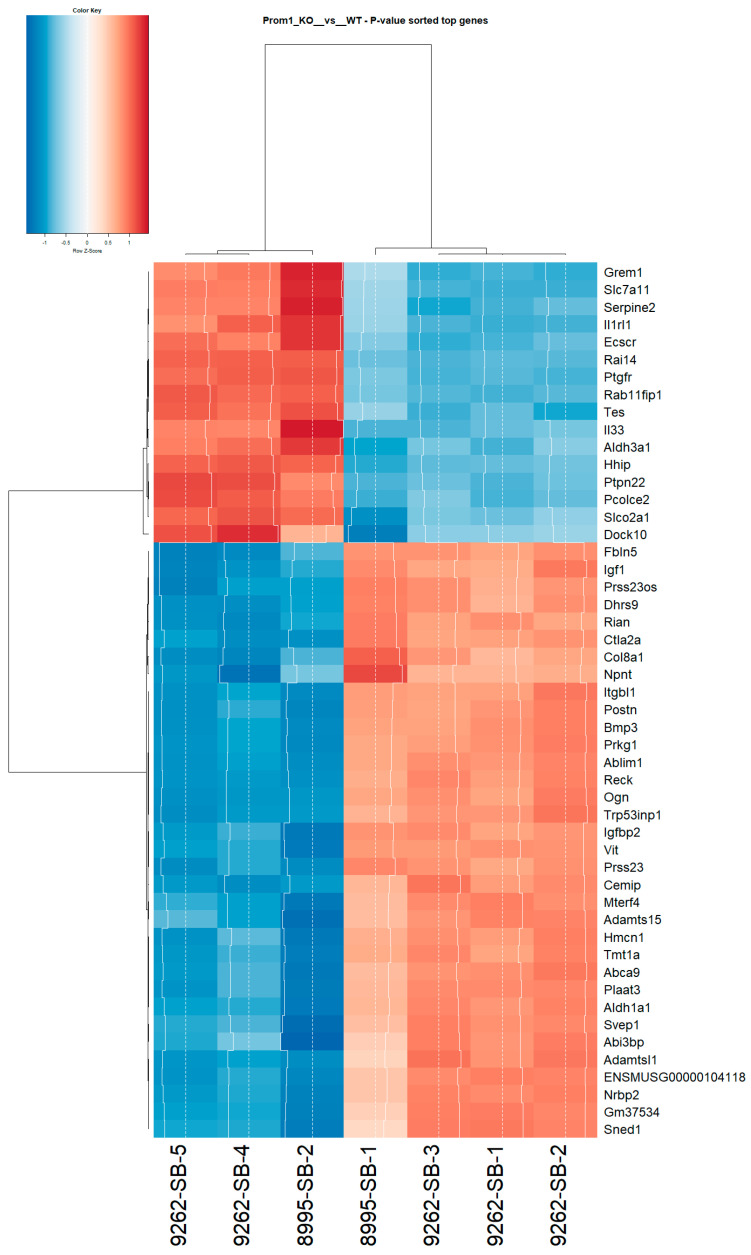
Heatmap of top DEGs between *Prom1*-KO and WT groups. Hierarchical clustering heatmap shows the top 50 DEGs ranked by *p*-value. Color intensity indicates relative expression levels, with red representing upregulation and blue representing downregulation. The clustering of samples is based on similarity in expression profiles. The color key (top left) shows the scale of normalized expression values. A HIGH-RES IMAGE WILL BE EMAILED AS AN ATTACHMENT.

**Figure 6 ijms-26-11539-f006:**
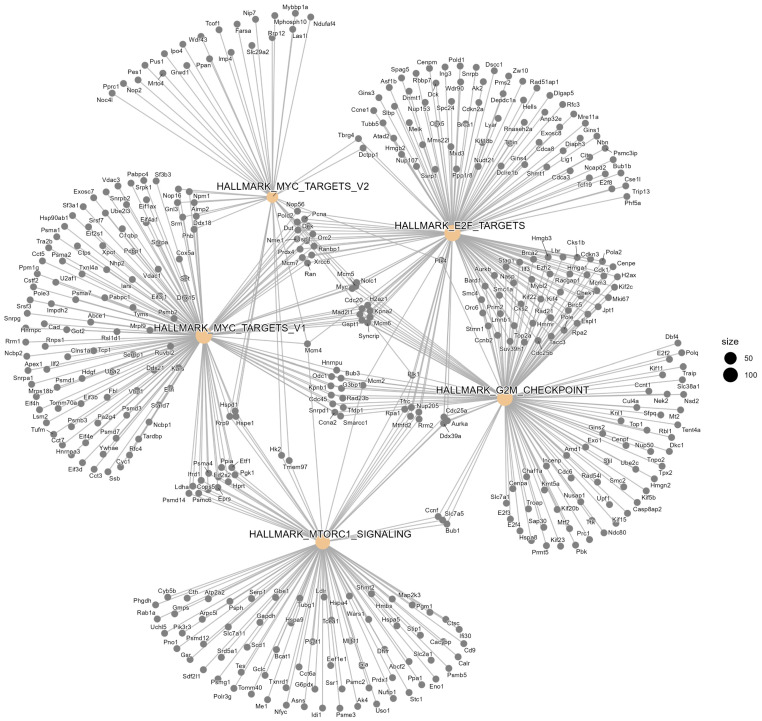
Network analysis of Hallmark gene sets reveals central regulatory pathways in RPE dysfunction. The network representation of hallmark gene sets illustrates the interconnections and relationships among key gene targets implicated in RPE pathology. Each node represents a gene set, and edges denote functional associations or co-regulation. Prominent nodes such as MYC_targets, E2F_targets, G2M_checkpoint, and MTORC1_signaling reflect central hubs in cell cycle regulation, proliferation, and stress signaling. Node size corresponds to connectivity, with larger nodes indicating higher degrees of interaction.

**Figure 7 ijms-26-11539-f007:**
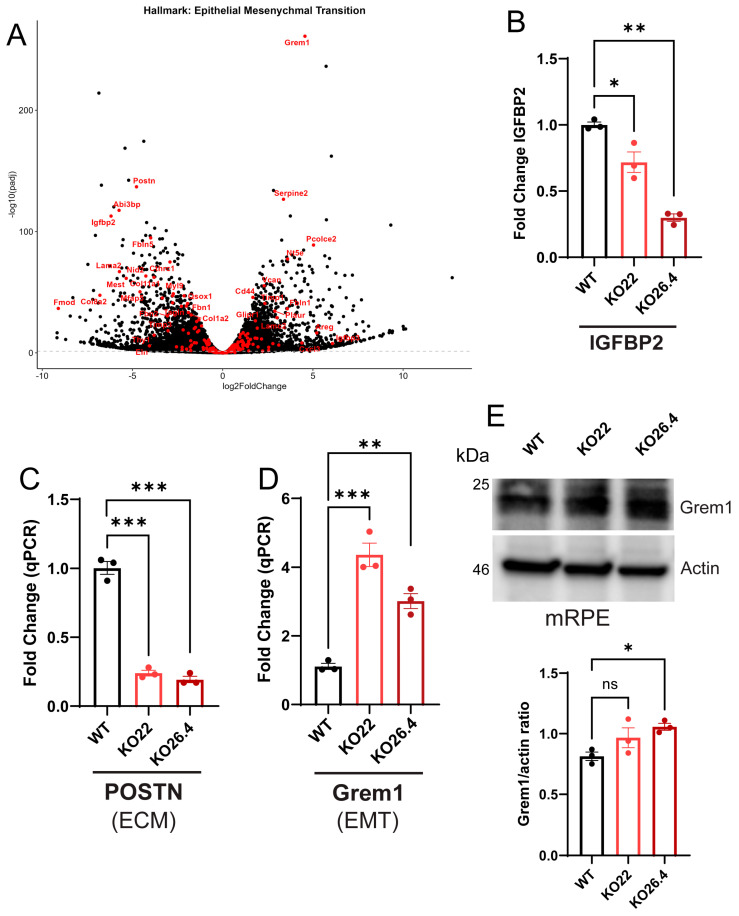
*Prom1* deficiency induces EMT-associated gene expression in mRPE cells. (**A**) Volcano plot of RNA-seq data highlighting differentially expressed genes associated with the hallmark EMT pathway. Genes with significant upregulation include *Grem1*, *Serpine 2*, and *Pcolce2*. Downregulated genes are *Postn*, *Abi3bp*, and *Igfbp2*. (**B**) Validation of *IGFBP2* gene expression by qPCR in KO22 and KO26.4 compared to WT (* *p* < 0.05, ** *p* < 0.01). (**C**) *POSTN* gene expression in KO lines relative to WT (*** *p* < 0.001). (**D**) *Grem1* gene expression in KO22 and KO26.4 compared to WT (** *p* < 0.01, *** *p* < 0.001). (**E**) Western blot analysis showing Grem1 protein levels in WT and Prom1-KO RPE cells, with Actin used as a loading control. Quantification of Grem1 and actin ratio (ns = not significant, * *p* < 0.05).

**Figure 8 ijms-26-11539-f008:**
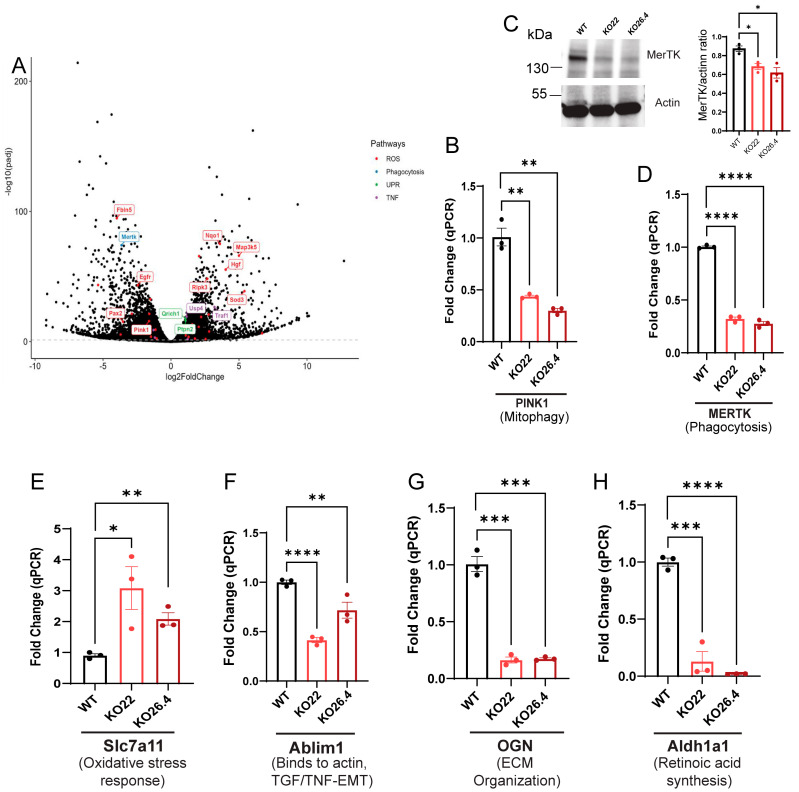
Transcriptomic and molecular validation of dysregulated pathways in *Prom1*-KO mRPE cells. Multi-panel analysis of gene expression and protein changes in *Prom1*-KO mRPE cells. (**A**) Volcano plot of bulk RNA-seq data showing significantly dysregulated genes in *Prom1*-KO mRPE cells. Pathways of interest are color-coded: general dysregulated genes (red), ROS (red), phagocytosis (blue), UPR (green), and TNF signaling (purple). (**B**) qPCR validation of *PINK1* in KO22 and KO26.4, compared to WT mRPE (** *p* < 0.01). (**C**) Western blot analysis of MerTk protein with *actin* as a loading control, quantification of MerTK and actin ratio (* *p* < 0.05), and (**D**) qPCR of *MerTK* gene in WT and Prom1-O samples (**** *p* < 0.0001). (**E**–**H**) qPCR validation of additional dysregulated genes: *Slc7a11*, *Ablim1*, *OGN*, and *Aldh1a1* fold changes with statistical significance (* *p* < 0.05, ** *p* < 0.01, *** *p* < 0.001, **** *p* < 0.0001).

**Table 1 ijms-26-11539-t001:** Primer sequences for qPCR of mouse genes.

Gene (Mouse)	Forward Primer	Reverse Primer
*Beta-actin*	CCTGGATAGCAACGTAGATGC	ACCTTCTACAATGACCTGGC
*Prom1*	AACATATGCGCGGGAGAG	CAGTTTCTGGGTCCCTTTGA
*Pink1*	CTGATCGAGGAGAAGCAGGC	GCCAATGGCTTGCCCTATGA
*Ogn*	CGCAGCTGGACTCACATGTT	TCTTTCTTGGTTGGTAATGATGCT
*Mertk*	TGGATACGTGCATCTGTCCG	GAGGAGCAGAGAATGGGCTG
*Grem1*	CTTCGCAGACCTGGAGACG	CAGGTTGTGGTGGGGACTG
*Slc7a11*	CAGGCATCTTCATCTCCCCC	GAGCAGTTCCACCCAGACTC
*Ablim1*	GAGGCCATCGGTCTGCTTC	GAAATGCTTGGTCTGCACCC
*Igfbp2*	CACAGGTGACACTGCAGACG	GAACACAGCCAGCTCCTTCA
*Aldh1a1*	TGAGCCTGTCACCTGTGTTC	CCTTCTTCCACGTGGCAGAT
*Postn*	ATGACAAGGTCCTGGCTCAC	CCCGCAGATAGCACCTTGAT

## Data Availability

The data supporting this study’s findings are available from the corresponding author upon reasonable request. The RNA sequence files have been deposited in NCBI under GEO accession ID GSE298665 (embargoed until publication).

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
