# Peer review of "Prominin-1 Regulates Retinal Pigment Epithelium Homeostasis: Transcriptomic Insights into Degenerative Mechanisms"

_ijms, 2025, doi:10.3390/ijms262311539_

Round 1

Reviewer 1 Report

Comments and Suggestions for Authors

This paper provides insights into the transcriptome of prom1-KO RPE and conducts preliminary validation of the pathways of interest. The following issues in this paper that need improvement are as follows:

  1. As a membrane protein, why does prom1 (CD133) exhibit significant cytoplasmic/nuclear staining instead of co-localizing with ZO-1 on the membrane?
  2. Are the volcano plots in Figures 4a, 7a, and 8a derived from the same dataset? Apart from the differences in the pathways of interest, what are the differences in the screening criteria for DEGs?
  3. Is the data cleaning process also one of the main conclusions of this paper? If not, it would be better to include it as a supplementary file.
  4. Please discuss the limitations of this paper.
  5. Low-quality image presentation: The scale bar in Figure 1 is too small to see clearly, and the text in most figures is also too small, which is not reader-friendly.
  6. In line 110, "CD1330" should be corrected to "CD133."

Author Response

Comment 1: As a membrane protein, why does prom1 (CD133) exhibit significant cytoplasmic/nuclear staining instead of co-localizing with ZO-1 on the membrane?

Response 1: We thank the reviewer for this excellent comment. Our earlier studies have shown cytosolic Prom1 localization in human RPE from donor explants (using immunocytochemistry) and mitochondrial localization in mouse RPE in situ (using immunogold electron microscopy). We have revised our discussion to clarify Prom1’s cytoplasmic and nuclear staining in mouse RPE flatmounts/retina sections instead of complete co-localization with ZO-1 (lines 430- 447). This revised portion is also added below.

Although Prom1 is a pentaspan membrane protein, its limited co-localization with ZO-1 in mRPE aligns with its known enrichment in cholesterol-rich, highly curved apical microdomains such as microvilli and cilia, regions that are spatially separate from the RPE tight junctions marked by ZO-1. This microdomain specificity, observed across epithelial and sensory cells, indicates that Prom1's role involves membrane curvature and protrusion dynamics rather than functioning as a junctional scaffold [9,65]. The prominent cytoplasmic and nuclear localization of Prom1 in mRPE in situ can be explained by several factors: 1) active vesicular trafficking and endosomal recycling, including its presence in microvillus-derived vesicles and intracellular vesicles termed intercellsomes [66]; 2) glycosylation-dependent epitope accessibility, where intracellular pools in the endoplasmic reticulum, Golgi, and the mitochondria may be more likely to be detected due to altered glycosylation patterns [65]; and 3) rare but documented nuclear trafficking in specific cell types, indicating biological plausibility for nuclear Prom1 signals [67]. Importantly, our published data confirm that this cytoplasmic distribution pattern is conserved in human RPE donor explants (using immunocytochemistry) and mouse RPE mitochondria in situ (using immunogold electron microscopy), further supporting the biological significance of these mechanisms, particularly as a cytoplasmic regulator of essential RPE functions such as autophagy [26,27].

Comment 2: Are the volcano plots in Figures 4a, 7a, and 8a derived from the same dataset? Apart from the differences in the pathways of interest, what are the differences in the screening criteria for DEGs?

Response 2: The volcano plots in 7a and 8a are derived from the same pairwise comparison. The DEG screening criteria are as follows:

  • abs(Log2FC) > 1
  • adjusted_pvalue <= 0.05 num

We have added the DEG screening criteria to the results section (lines 306-309)

Comment 3: Is the data cleaning process also one of the main conclusions of this paper? If not, it would be better to include it as a supplementary file.

Response 3: As requested, we have removed Fig. 3 from the original submission and included it as Supplemental Figure 1 (Fig. S1) in the revised manuscript.

Comment 4: Please discuss the limitations of this paper.

Response 4: We have added the limitations of this paper at the end of the discussion (lines 553-579). We have added this section below.

Our study has several limitations. First, while transcriptome profiling provides a comprehensive review of gene expression changes in Prom1-KO mRPE cells, it does not capture post-transcriptional regulation, protein turnover, and dynamic signaling events, all of which may profoundly influence disease phenotypes. Secondly, our study utilizes bulk RNA sequencing of Prom1-KO mRPE cells cultured in vitro, which does not fully reflect the in vivo RPE heterogeneity and spatial organization of the RPE in the central and peripheral regions. The in vitro system lacks the native tissue architecture and microenvironmental cues that influence RPE subpopulation dynamics, including partial EMT and stress adaptation. Third, although we integrate prior localization studies confirming that Prom1 is a cytoplasmic protein in both human and mouse RPE, our localization data in this study do not directly link the localization pattern to functional outcomes such as vesicular trafficking or signaling compartmentalization. Finally, while transcriptomic analyses provide valuable insights into primary-level perturbations in Prom1-KO mRPE cells, they reflect correlative changes in gene expression and do not fully account for dynamic regulatory mechanisms at the protein level. To specifically address this, we included protein-level validation for select markers, such as MerTK and GREM1; however, comprehensive protein profiling was limited by the lack of commercially available antibodies for several key targets. Future studies integrating proteomic approaches may help refine mechanistic interpretations related to mitochondrial quality, EMT, autophagy, and junctional integrity. The development of an RPE-specific inducible Prom1-Ko mouse model would be particularly valuable, as it would enable in vivo testing of whether Prom1 loss alone drives cell-autonomous RPE degeneration through similar transcriptomic changes that phenocopy key pathogenic mechanisms shared between aAMD and IRDs. In parallel, single-cell or spatial transcriptomic approaches applied to intact retinal tissue lacking Prom1 will be essential to validate and extend our findings, as well as to delineate context-dependent transcriptional responses across RPE subtypes within their native microenvironment.

Comment 5: Low-quality image presentation: The scale bar in Figure 1 is too small to see clearly, and the text in most figures is also too small, which is not reader-friendly.

Response 5: To improve the quality of image presentation, we have revised Fig. 1 from the original submission and split this figure into Figs. 1 and 2. Both figures contain enlarged scale bars and are in TIFF format with high resolution. To enhance clarity, Fig. 1A in the original submission is now Fig. 1, which shows Prom1 IHC of mouse RPE flatmounts. Fig. 1B from the original submission is now Fig. 2A. Please note that all panels in Fig. 2A include large-scale bars and are high-quality, large-format images to ensure readability.

We acknowledge that Fig. 6 (gene network graph) contains many gene names and maps, which are difficult to read as an embedded figure in the Word document. Increasing text size is not possible without label overlap in the gene-network graph (Fig. 6). We have included a high-resolution image in the revised version (Fig. 6). If viewers have trouble reading it, they may need to view it online (provided as a TIFF) and zoom in.

Comment 6: In line 110, "CD1330" should be corrected to "CD133."

Response: We have corrected this typo to CD133 in the revised manuscript (line 588).

Reviewer 2 Report

Comments and Suggestions for Authors

The manuscript by Huo and co-authors describes a timely investigation into the role of Prominin-1 (Prom1) in retinal pigment epithelium (RPE) homeostasis. The study effectively challenges the photoreceptor-centric view of Prom1-related retinal dystrophies by providing robust evidence of Prom1 expression in mouse RPE and demonstrating profound transcriptional and functional consequences of its loss. The use of bulk RNA-sequencing on CRISPR/Cas9-generated Prom1-KO mRPE cells is a major strength, revealing dysregulation in critical pathways such as mTORC1 signaling, mitophagy (PINK1), phagocytosis (MerTK), oxidative stress, and inflammation. The results are important and deserve publication.

Major concerns:

1). A central point of confusion arises from the apparent contradiction between the transcriptomic data (GSEA showing suppression of the Hallmark EMT pathway) and the authors' previous work and partial validation in this study (showing upregulation of individual EMT-promoting genes like Grem1). The authors propose this may represent an "incomplete or partial EMT." This is a plausible and interesting hypothesis, but it needs to be developed more thoroughly.

2). The suppression of the bile acid metabolism pathway is an intriguing finding, and the discussion on TUDCA/UDCA is relevant. However, this observation currently feels underdeveloped. The authors could expand the discussion to speculate on why this pathway might be suppressed and how it integrates with the other observed deficits (e.g., oxidative stress, inflammation). 

Minor comment:

3). Figure 3:  The resolution of the images is very low and should be enhanced. The same applies to Figure 5. 

Summarizing, I recommend major revision of the manuscript before acceptance.

Author Response

Comment 1:  A central point of confusion arises from the apparent contradiction between the transcriptomic data (GSEA showing suppression of the Hallmark EMT pathway) and the authors' previous work and partial validation in this study (showing upregulation of individual EMT-promoting genes like Grem1). The authors propose this may represent an "incomplete or partial EMT." This is a plausible and interesting hypothesis, but it needs to be developed more thoroughly.

Response 1: We thank the reviewer for this constructive comment, which has led to a more comprehensive and mechanistically cohesive Discussion section (lines 520-562). We have substantially expanded the Discussion section to elaborate on this point. We now frame this observation within the emerging concept of partial or hybrid EMT, as supported by recent literature (PMIDs: 32751632, 32671066, 40563412). We propose that Prom1-deficient cells upregulate individual EMT effectors but do not activate the full mesenchymal program, indicating a stalled transition, consistent with partial EMT. This framework reconciles the apparent contradiction and provides a more nuanced mechanistic interpretation that links transcriptomic and validation data. We have included the revised section below.

Our transcriptomic data from Prom1-KO mRPE cells reveal a paradoxical pattern: individual canonical EMT-promoting genes (such as GREM1) are upregulated, while Claudin family (Cldn1 and Cldn2) genes are downregulated, and Lhx2 is downregulated (Fig. S3). These changes indicate defects in the outer blood-retinal barrier and downregulation of the RPE differentiation program, leading to a less functional RPE. Yet, the global Hallmark EMT gene set is suppressed (using GSEA). At first glance, this seems contradictory. However, growing evidence in both epithelial biology and RPE research suggests that EMT is not a simple epithelial-mesenchymal switch, but rather a continuous spectrum of cell states, ranging from fully epithelial to hybrid or partial EMT, and ultimately to fully mesenchymal [81,82]. In the context of the RPE, a “partial EMT” phenotype is gaining attention, in which cells may lose epithelial polarity, begin to express mesenchymal markers, or acquire migratory potential, while retaining some epithelial features [80,83]. In Prom1-KO mRPE cells, the up-regulation of EMT-effector genes indicates the start of a phenotypic switch, but the overall suppression of the canonical EMT signature suggests that the complete mesenchymal program is not engaged. Therefore, we interpret this as a stalled or incomplete EMT- a state in which the cells embark on the transition path but remain locked in an intermediate state [80]. The partial EMT concept is highly relevant in degenerative RPE diseases (such as aAMD) rather than classical complete EMT seen in fibrosis or cancer. In aAMD, there is evidence for RPE cells acquiring migratory or ECM-remodeling characteristics while retaining epithelial identity [81]. We propose that Prom1 loss shifts cultured RPE cells toward a destabilized epithelial state by dismantling the epithelial program (i.e., reduced junctional proteins, ZO-1, and Claudins) but does not fully activate the mesenchymal signature (hence GSEA suppression). A partial EMT explains why Prom1-KO RPE cells exhibit junctional/ECM remodeling, yet do not fully adopt a fibroblast-like phenotype, consistent with an RPE cell continuum model in which cells move from a stable epithelial state toward mesenchymal phenotypes without completing differentiation. This hybrid state can be pathological because RPE cells lose their strict epithelial features but remain in place, leading to chronic degeneration rather than acute fibrosis. We therefore argue that the suppression of the Hallmark EMT gene set should not be seen as a lack of transitional change, but rather as a sign that the program is incomplete, consistent with the concept of hybrid EMT states in RPE. Future research will be needed to understand the functional effects of partial-EMT versus full-EMT states of the RPE in their natural location.

Comment 2: The suppression of the bile acid metabolism pathway is an intriguing finding, and the discussion on TUDCA/UDCA is relevant. However, this observation currently feels underdeveloped. The authors could expand the discussion to speculate on why this pathway might be suppressed and how it integrates with the other observed deficits (e.g., oxidative stress, inflammation). 

Response 2: The reviewer has raised a great point. We have modified the discussion section of the manuscript to speculate on why the bile acid signaling pathway may be suppressed and how it integrates with the other defects observed in Prom1-KO mouse RPE cells (lines 465-501).

Among our most intriguing findings is the robust suppression of the Hallmark Bile Acid metabolism pathway in Prom1-KO mRPE cells. Bile acids and their receptors (FXR and TGR5) are traditionally viewed from a hepatic-intestinal perspective; however, they also function as hormone-like immunometabolic signals that help limit inflammation, maintain epithelial barrier integrity, and regulate energy metabolism across various tissues [71].   [71]. Emerging evidence suggests that bile acid receptors in the retina/RPE function as potent immunometabolic regulators of inflammation, oxidative stress, epithelial barrier integrity, lipid homeostasis, and mitochondrial function. Several mechanisms could underlie the suppressed bile acid pathway signature in our model: 1) Loss of epithelial polarity and microdomain architecture (microvilli, membrane curvature) due to Prom1 deficiency likely perturbs bile acid receptor/transporter trafficking, thereby blunting its signaling; 2) Mitochondrial and lysosomal stress, including reduced PINK1 and mTORC1 activation caused by the loss of Prom1, can be mitigated or worsened by the FXR/TGR5 signaling axis due to their roles in mitochondrial biogenesis, quality control, and lipid metabolism [72]. Therefore, impaired bile acid metabolism could both be a cause and a consequence of mitochondrial dysfunction; 3) Inflammatory cross-talk, where reduced bile acid signaling removes an anti-inflammatory brake, potentially amplifying TNF-NF-kappaB signaling in Prom1-KO mRPE cells; 4) Gut-retina systemic-metabolic axis, as gut-derived bile acids can reach the eye and modulate oxidative/mitochondrial stress, suggesting altered metabolism or transport, could further sensitize Prom1-deficient RPE [73]. TUDCA/UDCA mitigates oxidative and ER stress, dampens pro-inflammatory cytokines, stabilizes barrier function, and promotes autophagy-mediated cytoprotection in RPE, suggesting that loss of bile acid signaling removes an endogenous pro-homeostatic brake in Prom1-deficient RPE and offers an actionable therapeutic axis [74,75]. Reduced MerTK-dependent phagocytosis and junctional/ECM remodeling increase structural strain, which bile acid signaling typically mitigates through blood retinal barrier support and FXR-mediated gene regulation [76,77]. The induction of SLC7A11 and ferroptosis-adaptive programs in Prom1-KO mRPE cells further suggests that bile acids can counter lipid peroxidative stress through lipid detoxification and crosstalk between peroxisomes and mitochondria. Our data may be among the first to link Prom1 loss to bile acids, highlighting therapeutic implications: restoring bile acid signaling via receptor agonists or strategies that augment endogenous bile acid metabolism could stabilize Prom1-deficient RPE and slow degeneration. These metabolite‑signaling considerations are consistent with broader lipid metabolic dysfunction observed in multi-omics RPE models and aAMD patient serum, reinforcing the clinical relevance of our metabolic signatures [78].  

Minor comment:

Comment 3: Figure 3:  The resolution of the images is very low and should be enhanced. The same applies to Figure 5. 

Response 3: To improve readability, Fig. 3 has been revised and now appears as supplemental Fig. S1.

Additionally, we have included a heatmap of the top 50 DEGs in Fig. 5, and the heatmap of the top 100 DEGs is provided as a supplementary file (Fig. S2). Due to the high resolution and large number of genes stacked tightly in the original heatmap, there is no way to improve legibility, unless the viewer downloads and views an enhanced version of the image. To address this, we have included the original Fig. 5 as a supplementary file. The revised version of Fig. 5 now contains only the top 50 genes, as it is more readable.

We recognize that Fig. 6 (gene network graph) contains many gene names and maps, which are hard to read when embedded in the Word document. There is no way to increase text sizing without label colliding in the gene-network graph (Fig. 6). We have included a high-resolution image in the revised version (Fig. 6). If viewers have difficulty reading, they may need to view the image online (provided as a TIFF format) and zoom in.

Summarizing, I recommend major revision of the manuscript before acceptance.

We have addressed the reviewers' comments and provided a point-by-point response to each.

Round 2

Reviewer 1 Report

Comments and Suggestions for Authors

The issues have been solved.

Reviewer 2 Report

Comments and Suggestions for Authors

The revised version of the manuscript was significantly improved by the authors. The comments were addressed properly. I recommend acceptance of the manuscript for publication in the revised form.